# Atomic Simulations of Packing Structures, Local Stress and Mechanical Properties for One Silicon Lattice with Single Vacancy on Heating

**DOI:** 10.3390/ma14113127

**Published:** 2021-06-07

**Authors:** Feng Dai, Dandan Zhao, Lin Zhang

**Affiliations:** 1Key Laboratory for Anisotropy and Texture of Materials Ministry of Education, Northeastern University, Shenyang 110819, China; 1610115@stu.neu.edu.cn; 2State Key Laboratory of Rolling and Automation, Northeastern University, Shenyang 110819, China; 1810147@stu.neu.edu.cn; 3School of Materials Science and Engineering, Northeastern University, Shenyang 110819, China

**Keywords:** silicon, defects, molecular dynamics, mechanical properties

## Abstract

The effect of vacancy defects on the structure and mechanical properties of semiconductor silicon materials is of great significance to the development of novel microelectronic materials and the processes of semiconductor sensors. In this paper, molecular dynamics is used to simulate the atomic packing structure, local stress evolution and mechanical properties of a perfect lattice and silicon crystal with a single vacancy defect on heating. In addition, their influences on the change in Young’s modulus are also analyzed. The atomic simulations show that in the lower temperature range, the existence of vacancy defects reduces the Young’s modulus of the silicon lattice. With the increase in temperature, the local stress distribution of the atoms in the lattice changes due to the migration of the vacancy. At high temperatures, the Young’s modulus of the silicon lattice changes in anisotropic patterns. For the lattice with the vacancy, when the temperature is higher than 1500 K, the number and degree of distortion in the lattice increase significantly, the obvious single vacancy and its adjacent atoms contracting inward structure disappears and the defects in the lattice present complex patterns. By applying uniaxial tensile force, it can be found that the temperature has a significant effect on the elasticity–plasticity behaviors of the Si lattice with the vacancy.

## 1. Introduction

With the continuous reduction in the chip size of microelectronic devices with high-cost performance, higher requirements have been proposed for the processing of materials used in these devices. Since the concept of micromachining based on silicon materials was proposed in the 1970s, it has been the key to technological breakthroughs of the microelectromechanical system (MEMS) for the fabrication of micromovable structures based on silicon materials employed by micromachining technology, which are compatible with integrated circuits, for manufacturing microsystems [1]. Especially in the recent decades, the rapid developments of the internet, unmanned driving, wise medical and smart robots have provided unlimited possibilities for applying MEMS sensors in microelectronic devices [2,3,4,5,6,7,8,9].

MEMS is a miniaturized device or a combination of these devices. This system integrates electronics, machinery, optics and other functions to achieve intelligent effects in minimal space. In order to achieve these effects, a breakthrough in the manufacturing technology is required. When the size of a MEMS device is reduced to a certain range, they show many physical phenomena that are different from those in macroscopic systems. For example, the tensile testing data present distinguished differences for the millimeter and micrometer samples [10]. When the size or the spacing between units is within 1 μm or less, the processing technology for these three-dimensional devices is called micromachinery. As micromachining was originally developed from silicon microelectronics processing technology, it is also called silicon-based micromachining [11,12].

As the most important semiconductor material, crystalline silicon has a diamond structure and covalent bonds among atoms. It has excellent mechanical properties, such as high strength and high hardness, as well as good thermal conductivity. Meanwhile, it also has excellent characteristics in light, heat, electricity, magnetic and other properties, and thus can be integrated into capacitive sensors, thermoelectric light detectors, hot gas pressure sensors, magnetometers and photoelectric monitors [13,14,15,16,17,18]. In these MEMS devices, there are many Si wafers, which are flat or have different shapes, including beams, bridges and probe arms. They can be used as light-conducting devices for optical communication, grating devices, micronozzles, microvalves, pumps, micropipes, etc. [19,20]. Moreover, non-linear cantilever, small arc parts, etc., appear in MEMS microsensors, such as tactile sensors on the robot manipulators and chemical reaction sensors [21]. These silicon wafers with different shapes and structures need to be plastic processed. However, the fracture strength of silicon is very low. Various observation approaches give values of tensile fracture strength in the range of 3 to 7 GPa [22,23]. In addition, early loading experiments with a bearing ball on mirror-polished silicon wafers yield an average fracture stress of 2.8 GPa and a maximum value of 6.9 GPa [24]. The transition from elasticity to plasticity occurs only when the temperature exceeds 790 K. When the temperature is above 920 K, the plastic forming becomes easy [25,26]. At the end of the last century, J. Frühauf et al. [25,26] proposed a laser technology to make materials plastically bend. When laser beams scan the silicon-based surface, plastic deformation occurs owing to the fact that non-uniform temperature field generates thermal stress in the materials [27]. The defects in the silicon bulk greatly affect the forming process. Vacancies and interstitial atoms are the two most important primary point defects in silicon single crystals [28]. Some silicon atoms can remove the binding of the surrounding atoms and jump away from the equilibrium position to form vacancies under specific conditions. Subsequently, the atoms entering the lattice spaces become interstitial atoms [29]. These vacancies can trap the carriers in the silicon crystal by generating deep energy levels, resulting in a decrease in the number of carriers, and affect the performance of semiconductor devices [30]. This kind of defect is not only related to the formation of other forms of defects, but also controls the diffusion of interstitial atoms in semiconductors. At the same time, the existence of the vacancies provides greater possibilities for material deformation [31]. Therefore, it has always attracted attention from experimental and theoretical researchers [32,33]. In the laser bending process, the silicon absorbs the energy from the laser irradiation, so as to produce an uneven transient temperature field in the material matrix. When the temperature of the material surface rapidly increases, the heat generated by the laser diffuses into the material. This additional energy results in changes in the configurations and numbers of defects inside the matrix, and eventually changes the mechanical properties of the crystal. As the heating process is only controlled by the macrotemperature field in the experiment, it poses great difficulties in observing and measuring the changes in the microstructure and mechanical properties in the matrix containing vacancy defects under the conditions of rapid heating by high-energy laser irradiation. Thus, computer simulation based on the empirical potential, such as molecular dynamics, has become a powerful tool to study atomic movements, packing evolution and stress distribution.

The Stillinger–Weber (SW) potential [34] is composed of two-body and three-body potential functions. It can give the strain energy in a few potential parameters and can be used to describe many kinds of defects in silicon. At present, the SW potential function has been successfully applied in studying vacancy defects [35]. In this paper, the molecular dynamics method based on the SW empirical potential [34] of the interaction among atoms in silicon crystalline was used to simulate the heating process of silicon crystal with single vacancy and analyze the packing structures, atomic localization, stress and Young’s modulus of the crystal change with temperature.

## 2. Simulation and Method

The interaction among silicon atoms is described by the SW potential. This potential contains two-body and three-body parts [34] given by the following formula:(1)$$U=U_{2}\left(r_{ij}\right)+U_{3}\left(r_{i},r_{j},r_{k}\right)$$

Here, *U*_2_ and *U*_3_ represent the potential of the two-body and three-body parts, respectively, and r is the distance between atoms. *U*_2_ and *U*_3_ are expressed as:(2)$$U_{2}\left(r_{ij}\right)=\varepsilon f_{2}\left(\frac{r_{ij}}{\sigma }\right)$$
(3)$$U_{3}\left(r_{i},r_{j},r_{k}\right)=\varepsilon f_{3}\left(\frac{r_{i}}{\sigma },\frac{r_{j}}{\sigma },\frac{r_{k}}{\sigma }\right)$$

The potential well constant ε, the equilibrium constant σ and the atomic mass are dimensionless quantities. *f*_2_ and *f*_3_ are dimensionless two-body and three-body potential, and their forms are as follows:(4)$$f_{2}\left(r\right)=\{\begin{array}{c} A\left(Br^{-p}-r^{-q}\right)exp\left(\frac{1}{r-a}\right),\;\;\;\;r<a\\ \;\;0,\;\;\;\;\;\;\;\;\;\;\;\;\;\;\;\;\;\;\;\;\;\;\;\;\;\;\;\;\;\;\;\;\;\;\;\;\;\;\;\;\;\;\;\;r\geq a \end{array}$$
(5)$$f_{3}=\{\begin{array}{c} (r_{i},r_{j},r_{k})\;=\;h(r_{ij},r_{ik},\theta_{jik})\;+\;h(r_{ji},r_{jk},\theta_{ijk})\;+\;h(r_{ki},r_{kj},\theta_{ikj}),\;\;\;\;r<a\\ \;\;0,\;\;\;\;\;\;\;\;\;\;\;\;\;\;\;\;\;\;\;\;\;\;\;\;\;\;\;\;\;\;\;\;\;\;\;\;\;\;\;\;\;\;\;\;\;\;\;\;\;\;\;\;\;\;\;\;\;\;\;\;\;\;\;\;\;\;\;\;\;\;\;\;\;\;\;\;\;\;\;\;\;\;\;\;\;\;\;\;\;\;\;\;\;\;\;\;\;\;\;\;\;\;\;\;\;\;\;\;r\geq a \end{array}$$
where *θ_jik_* is the angle between *r_j_* and *r_k_* subtended at vertex *i*, etc. The parameter a is chosen so that the potential is truncated between the first-nearest-neighbor and the second-nearest-neighbor distances. The function h belongs to a two-parameter family (*λ*, γ > 0). Provided that both *r_ij_* and *r_ik_* are less than the cut off *a*, it has the following form:(6)$$h_{3}\left(r_{ij},r_{ik},\theta_{jik}\right)=\lambda exp\left[\frac{\gamma }{r_{ij}-a}+\frac{\gamma }{r_{ik}-a}\right]\times {\left(\frac{1}{3}+\cos\theta_{jik}\right)}^{2}$$

*σ*, *ε*, *A*, *B*, *λ*, *γ*, *p*, *q* and *a* are adjustment parameters. The parameters in the empirical potential are obtained from experiments and calculations based on quantum mechanics. Table 1 shows the relevant parameters of the potential.

This paper used the General Utility Lattice Program (GULP) software (version 4.5) developed by Julian D. Gale of Curtin University, Australia. [36,37]. Each time step corresponds to a real time of 1.0 × 10^−15^ s. Initially, two structural models were constructed. As shown in Figure 1a, the cell size of the perfect silicon block is 3a × 3a × 3a (a refers to the lattice constant; a = 5.431 Å), containing 216 Si atoms. Figure 1b shows the cell size of the silicon with a vacancy defect 3a × 3a × 3a, containing 215 atoms, where the red dashed ellipse in the figure indicates the vacancy. Periodic boundary conditions are applied along the three directions of [100], [010] and [001].

In the calculations, the constructed initial structure is firstly subjected to structural relaxation at 300 K, and then the temperature is increased at intervals of 100 K. During the heating process, the last-step structure obtained by structural relaxation at each temperature is used as the initial structure at the next temperature. The stiffness of a material is expressed by the Young’s modulus E, which can be used to describe the ability of a solid material to resist deformation. For the lattice system, the elastic constant matrix is obtained by calculating the second derivative of the strain of the interaction energy between atoms, and then the Young’s modulus along the three axial directions can be obtained from the reciprocal of the elastic flexibility coefficient.

In the present simulations, atomic local stress was introduced to analyze the stress of the small volume element occupied by one atom. The stress tensor of each atom is given by the following formula:(7)$$\sigma_{i}{}^{ab}=\frac{1}{V_{i}}\underset{j\neq i}{\sum }\frac{\partial E_{i}}{\partial r_{ij}}\frac{r_{ij}{}^{a}r_{ij}{}^{b}}{r_{ij}}$$
where *r_ij_^a^* and *r_ij_^b^* are the Cartesian components of the vector, and *r_ij_* is their modulus. *V_i_* is the volume of atoms, and *E_i_* is the energy of the *i*th atom. The relationship between the hydrostatic pressure of each atom *P_i_* and *σ**_i_* is given by the following formula:(8)$$P_{i}=\frac{1}{3}\left(\sigma_{i}{}^{xx}+\sigma_{i}{}^{yy}+\sigma_{i}{}^{zz}\right)$$

In applying the strain for the Si lattice, the cell length along the [100] axial direction of the Si lattice is increased at an increment of 1% of the initial length at 4000 steps, corresponding to a strain rate of 0.0371 Å/ps. Stress is derived from the sum of the hydrostatic pressure of these atoms.

## 3. Results and Discussion

Figure 2 shows the variations of the potential energy per atom with temperature before melting for perfect and single-vacancy lattices. As shown in the figure, at room temperature, the potential energy with a single vacancy was higher than that of the perfect lattice. The higher energy is due to the fact that the number of coordination atoms of the atom near the vacancy was lower than that of the atoms in a perfect lattice. Before melting, the energy of the perfect lattice increased linearly with the increase in the temperature, which is due to the intensification of the thermal vibrations of the atoms around their lattice positions. For the lattice containing a single vacancy, the average potential energy increased linearly below 1800 K. When the temperature was higher than 1900 K, there was an apparent decrease in potential energy, and then a significant increase occurred at 2100 K. For a vacancy in the bulk, its neighboring atoms had fewer coordination numbers than those far from the vacancy. With the increase in the temperature, the neighboring atoms will show more obvious thermal motion, and some of them need only a small amount of extra energy to move, which can be provided at high temperatures. The motion resulted in changes in the distances between the atoms. Correspondingly, the potential energy was changed. Therefore, we found that the apparent energy changes at high temperatures. When the temperature increased by 20 K, it can be seen on the bottom right of this figure that the changes in the potential energy became smooth in the temperature range of 1200 to 2300 K. This suggests that the heating process can significantly affect the motion of the atoms near the vacancy.

Figure 3a,b show the variation of the Young’s modulus values along the three crystal directions, [100], [010] and [001], with the temperature for the two cases. When calculating the Young’s modulus, we selected the structure corresponding to the lowest potential energy value in the last 10,000 steps of the simulation time steps at each temperature. As shown in Figure 3a, when the temperature was lower than 1200 K, the Young’s modulus values of the perfect lattice along the three crystal directions decreased in a nearly linear pattern. This suggests that in this temperature regime, the crystal shows obvious elasticity. With the increase in temperature, the thermal movement of atoms intensified, and the distance between atoms changed correspondingly. As the elastic constant matrix is derived from the second derivatives of the strain of the interaction energy between the atoms, the elasticity decreased in elastic limits [38]. Above 1300 K, the Young’s modulus value along the [100] direction decreased with a larger slope. When the temperature was higher than 1600 K, the values of Young’s modulus were distributed in scattered points in a high-temperature regime. Following a small fluctuation, there was a sharp decrease at 1600 K along the [010] direction, and then the Young’s modulus fluctuated. At 1400 K, the maximum value of the Young’s modulus was observed along the [001] direction. With the increase in temperature, a downward behavior was observed, followed by a fluctuation at higher temperatures. In order to better observe the scattering behavior of the Young’s modulus, Figure 4(c_1_–c_3_,d_1_–d_3_) illustrate the Young’s modulus in the temperature range of 1200–2300 K with a temperature increase of 20 K. As illustrated in these figures, the scattering behavior was more obvious. This suggests that when the temperature is higher than 1200 K, the distance between the atoms increases and exceed the limit of elasticity. Therefore, by using the method of calculating the second derivative of the strain of the interaction energy between atoms to obtain the elastic constant matrix, the scattering behavior occurred. This implies that as a typical brittle material, crystalline silicon has a certain brittle–ductile transition temperature. When it is higher than this temperature, the silicon is capable of plastic deformation, and the elastic modulus is relatively small. When it is lower than this temperature, the elastic modulus of silicon is relatively large and prone to brittle fracture, which makes plastic forming difficult at low temperatures [39,40]. In the present simulations, the temperature range for difficult plastic deformation was lower than 1200 K, and the transformation to plasticity occurred in the temperature range of 1200–1500 K. When the temperature was higher than 1600 K, it was beneficial to plastic forming. It can be observed that different elastic–plastic temperature ranges from the calculation results were observed in the experiments [25,26], though the calculated temperature point value was higher than the results given in the literature. Accounting for the fact that the calculated melting point Tm was 2396 K, this point was higher than the experimental melting point of 1687 K [41]. The relative temperature of the calculated elastic-plasticity transition point was 0.50 Tm, which is close to the experimental value. Melting is a transition from order to disorder caused by the elastic instability of lattice. In the actual melting process, the initial liquid phase is generally formed from the surface or interface, etc., and then expands the entire solid [42,43,44]. As there are no defects in the perfect lattice, the simulated melting point is higher than the experimental melting point.

For the vacancy case, the Young’s modulus presented a different behavior from its perfect counterpart. As shown in Figure 3b, at room temperature, the Young’s modulus was slightly lower than that of the perfect lattice. When the temperature was lower than 1100 K, the modulus along the [100] direction decreased with a small oscillation. Then, the decrease with a large slope appeared, and at 2100 K, a low modulus value was observed. Below 1500 K, large modulus values appeared along the [010] and [001] directions. As the temperature increased further, the decrease in the modulus along the [010] direction was oscillatory. When the temperature was higher than 1900 K, it exhibited an accelerated decline behavior.

Figure 4 shows the atomic packing structures and the distribution of the pressure on the atoms for the vacancy lattice, as well as enlarged images of the packing structures near the vacancy, and the positive/negative pressures of these atoms neighboring the vacancy at 300, 400, 700, 1200 and 1500 K. The red color in this figure suggests that the atom undergoes compression, and the pressure is positive. The dark red color indicates that the atom is under pressure greater than 0.01 GPa, and the light red color less than 0.01 GPa. Blue atoms are stretched, and the pressure is negative. The dark blue color implies that the atoms are under pressure less than −0.01 GPa, and the light blue color greater than −0.01 GPa. The four nearest neighbors of the vacancy in the initial structure are labeled as 132, 135, 110 and 111, respectively. As shown in Figure 4(a_2_), at 300 K, these four atoms were compressed toward the vacancy. In the following, we selected the atom of 132. The distance between the two neighboring atoms (labeled as 130 and 129) was 2.42 and 2.52 Å. In them, the 130th atom was compressed, whereas the 129th atom was stretched. At the same time, the other three nearest neighbor atoms of 135, 110 and 111 of the vacancy and their nearest neighbors were also subject to tension or compression, resulting in changes in the distances between them. The four nearest neighbor atoms of the vacancy underwent greater pressure, and obvious shrinkage, as shown in the packing structures, appeared.

With the increase in the temperature, thermal movements of the atoms intensified, and the migration of the vacancy occurred. As shown in Figure 4(b_2_), at 400 K, the four nearest neighbors of the vacancy were the atoms of 135, 138, 201 and 208. All four atoms shrank toward the vacancy. Figure 4(b_3_) shows the pressure on the neighboring atoms of these four atoms. The atoms of 132 and 110, which were adjacent to 135, were compressed. Among them, 132 was subjected to great pressure, and 110 low pressure. The atoms of 133 and 140, which were adjacent to atom 138, were also compressed, of which the pressure on 133 was greater, and the pressure on 140 was lower. At the same time, 132 being adjacent to 201 was stretched, while 205 compressed. The atoms of 181 and 212, being adjacent to 208, were compressed, of which 181 was under less pressure, and 212 was under greater pressure. With the migration of the vacancy, there were the changes in the pressure on the neighboring atoms of the vacancy as well as the adjacent atoms, which affected the elastic properties of the entire matrix. As shown in Figure 3b, the Young’s modulus at 400 K was slightly higher than that at 300 K. As the temperature reached 700 K, Figure 4(c_2_) shows that the four nearest neighbor atoms of the vacancy changed back to the atoms of 132, 135, 110 and 111, whereas they had larger oscillations compared to those at lower temperatures. Meanwhile, the pressure of these atoms and their adjacent atoms was different to that at low temperatures. The atom of 131 was compressed, while the nearest neighbor of atom 135 became 201, which was stretched. The nearest neighbors of 111 were the two atoms of 177 and 200, of which 177 was compressed, and atom 200 was pulled. The nearest neighbors of 110 were 136 and 106, of which 136 was under pressure and 106 tension. Among these atoms, the pressure of the four nearest neighbors of the vacancy was significantly higher than that of the second neighbors and their surrounding neighbors. Then, the four nearest neighbors were still packed into a contraction structure toward the vacancy. Therefore, the corresponding Young’s modulus showed a significant decrease, as shown in Figure 3b. At 1200 K, the nearest neighbors of the vacancy became the atoms of 40, 43, 106 and 113. Then, the Young’s modulus along the [100] direction increased, while the Young’s modulus decreased along the [010] or [001] direction. This indicates that the elasticity of the matrix has an obvious anisotropy, and there is a transformation to plasticity. When the temperature was higher than 1200 K, the irregular and violent oscillation of the Young’s modulus suggested that the elasticity had disappeared. In the atomic packing structure at 1500 K, as shown in Figure 4(e_1_), the single-vacancy image existing at low temperatures disappeared, and was replaced by long loop defects. Most of the atoms making up the defect ring were compressed. Only the atoms of 113 and 94 were stretched, but their two nearest neighbors were also compressed. At the same time, the pressure values of the defect ring atoms were obviously higher than those of the nearest neighbor atoms of these ring atoms. Due to the effect of increasing temperature as well as defects, large distortions of lattice points occurred in the matrix.

Figure 5 shows the variation of the stress with the applied strain for a prefect silicon bulk at room temperature and the silicon lattice with one vacancy at 300, 1200 and 1300 K under increasing strain along the [100] direction. As shown in Figure 5a, for the Si perfect bulk, the change in stress with strain presented a smooth curve, which is a typical power-strengthening model curve and indicates the low plasticity of the bulk. When there was a vacancy defect in the silicon lattice, the stress–strain curve showed the characteristics of plastic materials. At room temperature, when the strain was small, the stress increased with a large slope. When the strain reached 0.01, the corresponding stress was 1.51 GPa, and then the growth slope gradually decreased, indicating that the Si lattice with vacancies exceeded the elastic strain limit and began microplastic deformation. In the microplastic deformation stage, most of the Si atoms in the lattice were shown in blue, indicating that they were under a uniformly distributed positive pressure. However, the positive pressure of the atoms adjacent to the vacancy was significantly higher than that of the surrounding atoms, which was yellow-green. As the strain increased to 0.13, the nearest neighbor atoms of the vacancy were significantly stretched along the [−1 0 0] direction, and the stress was 10.32 GPa. The stage from B point to C point in Figure 5a indicates that the lattice still continued to be elongated under the condition of decreasing stress, and the C point is the yield point. Image C on the right shows that the pressure of atoms was no longer uniformly distributed, and there were many large pressure regions in the lattice. As the strain continued to increase, the stress value increased again, and the ability of the Si lattice to resist deformation increased obviously under positive pressure, which entered the first stage of plastic deformation strengthening. When the strain was 0.23, the steep stress drop point D appeared. By observing the stress distribution, it was found that the vacancy was severely stretched into a cluster of vacancy rings, which showed that the yield platform here was caused by the expansion of the defects. As the strain continued to increase to 0.26, the maximum stress that the lattice could withstand was 14.91 GPa, reaching the tensile strength of the lattice, until the lattice was completely broken at point F. As shown in image F, the fracture was in the axial direction where the vacancy ring was located, and it was in the form of a cluster of micropores.

The stress–strain curves of the Si lattice were stretched at 1200 K, as shown in Figure 5b in the fluctuating plastic deformation behavior. When the strain was 0.01 and the stress was 1.24 GPa, the slope of the curve was the largest. As shown in the pressure image A on the right, the distribution of pressure on the atoms in the lattice no longer had a uniform distribution at room temperature. At this temperature, positive and negative pressure regions containing different atom numbers appeared in the lattice, which showed obvious plastic deformation. As the strain increased, the stress also increased until a slightly decreasing gentle yield platform appeared when the strain was 0.1. Then, the stress value was 7.42 GPa. In image B corresponding to the stress value, the single vacancy is no longer visible. Then, the curve grew in an oscillating manner until point C, reaching the maximum stress of 12.37 GPa, at which point the strain was 0.21. As shown in image C, fracture patterns with obvious defect aggregation appeared in the lattice. It can be seen that temperature significantly changed the elastoplastic properties of the Si lattice. When the temperature increased to 1300 K, as the strain increased, the stress presented different behaviors to that at a lower temperature. Following the increase in the stress with a larger slope, when the strain value was greater than 0.05, the stress growth became slower. Subsequently, the stress quickly increased again, and at the strain of 0.12, a high stress value appeared. In pressure distribution image A, at the strain of 0.01, it can be seen that the atoms in the lattice were subjected to uneven positive and negative pressures. When the strain increased to 0.12, the pressure distribution image corresponding to the stress point B showed a small range of atomic distances to reduce the aggregation area, until the strain reached the maximum stress of 0.18. Then, as the strain increased, the crystal lattice began to fracture.

Figure 6a shows the changes in yield strength and tensile strength of silicon lattices containing one vacancy at different temperatures and a temperature increase of 100 K below 1200 K during the tensile process. For the perfect lattice in the temperature regime of the present simulations, there were no apparent yield stages, and the fracture occurred only when the distance between atoms exceeded the cut-off radius of the potential function. As shown in this figure, the tensile strength of the perfect lattice was apparently higher than that of lattice with the vacancy. As shown in this figure, the curve of yield strength with the temperature decreased in oscillation mode, and the lowest value appeared at 900 K. This is due to the fact that during the tensile process, the increase in temperature affected the migration of vacancies and had a significant impact on the stress distribution of atoms around the vacancy. The maximum tensile strength of the lattice with the vacancy was 300 K, and it decreased significantly at 400 K. With the increase in temperature to 1000 K, the slope decreased until 1100 K. Following an increase, the tensile strength decreased. In order to better observe the changes in the yield strength and tensile strength of silicon blocks containing the vacancy as a function of the temperature, Figure 6b shows the yield strength and tensile strength varying with a temperature increase of 20 K.

## 4. Conclusions

In this paper, atomic simulations within the empirical potential framework were performed on the changes in the packing structure, local stress and mechanical behaviors of the perfect and single-vacancy silicon lattices with increasing temperature. The simulation results show that the silicon lattice has obvious elasticity, elasticity–plasticity transition and plasticity temperature ranges. The existence of the vacancy significantly reduces the elasticity–plasticity transition temperature and greatly affects the mechanical properties of the silicon lattice. In the elastic temperature range, the Young’s modulus with the vacancy is lower than that of a perfect lattice. With the increase in temperature, the vacancy migrates, which leads to the change in stress distribution in the region of its nearest neighbor atoms. As the distance between atoms increases largely, the atoms leave their equilibrium positions, and the lattice loses its elasticity, causing the silicon with the vacancies to show plasticity; correspondingly, the Young’s modulus of the lattice with a vacancy in the plastic temperature range significantly fluctuates. In the temperature range of elasticity–plasticity transition, the Young’s modulus of the lattice is anisotropic. At 1500 K, the single vacancy and shrinkage of its neighboring atoms disappear obviously, and the crystal lattice is distorted greatly. With a further increase in temperature, the defects in the crystal lattice show a complex pattern, and the number of atoms under high pressure increases obviously. The temperature significantly affects the mechanical behavior of the silicon lattice with a vacancy. The elongation of the lattice decreases during the heating process. In the temperature range in which plastic deformation occurs, the tensile strength of the lattice decreases. Due to the existence of the vacancy, the elastic and plastic behaviors present apparent differences with the increase in temperature.

## Figures and Tables

**Figure 1 materials-14-03127-f001:**
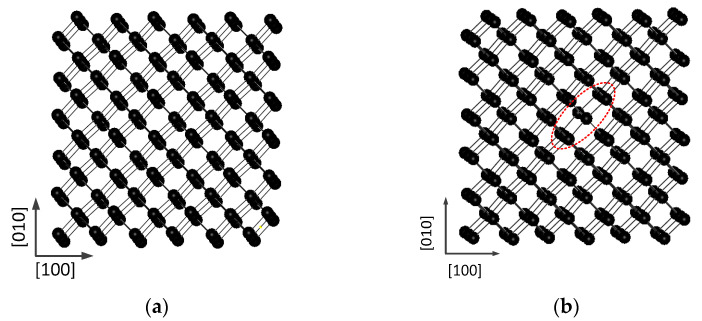
(**a**) Perfect Si lattice; (**b**) Si lattice with one vacancy.

**Figure 2 materials-14-03127-f002:**
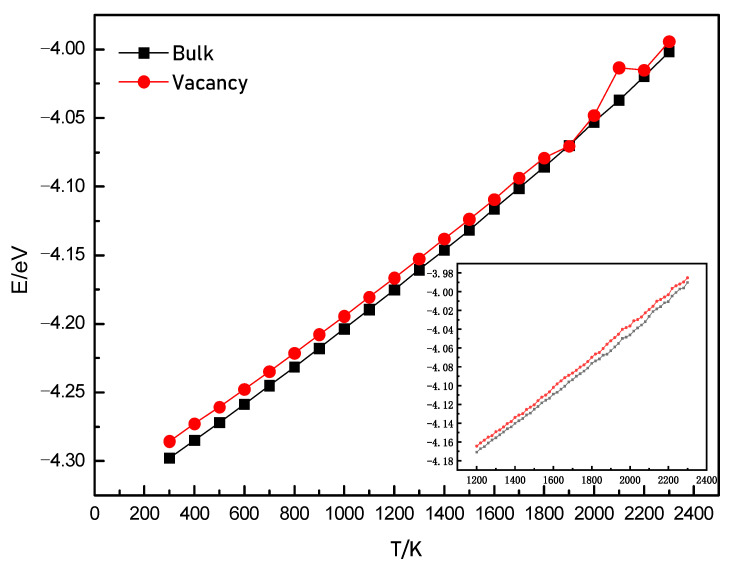
The average potential energy per atom varying with the temperature.

**Figure 3 materials-14-03127-f003:**
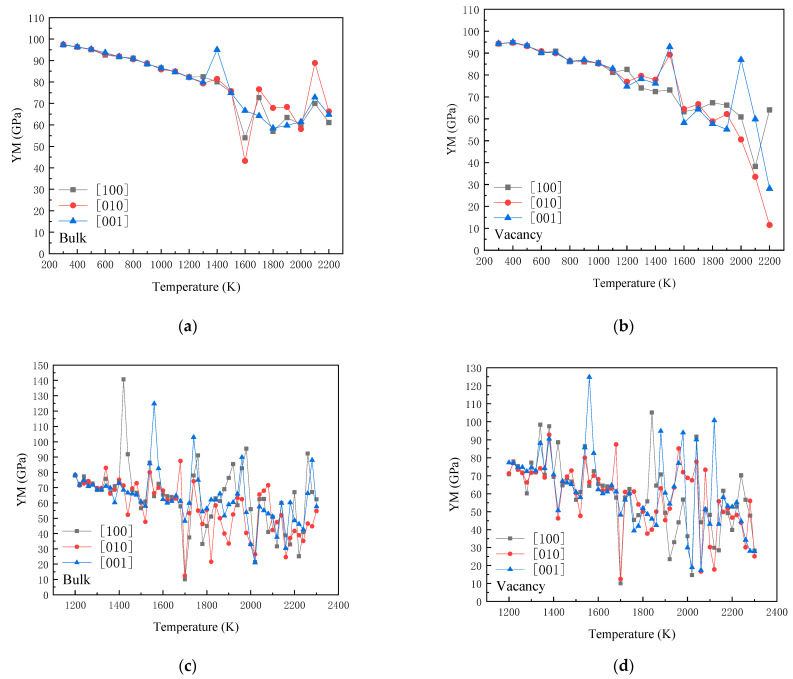
Variation of Young’s modulus with the temperature: (**a**) perfect lattice with a temperature increase of 100 K; (**b**) lattice with one vacancy with a temperature increase of 100 K; (**c**) perfect lattice with a temperature increase of 20 K; (**d**) lattice with one vacancy with a temperature increase of 20 K.

**Figure 4 materials-14-03127-f004:**
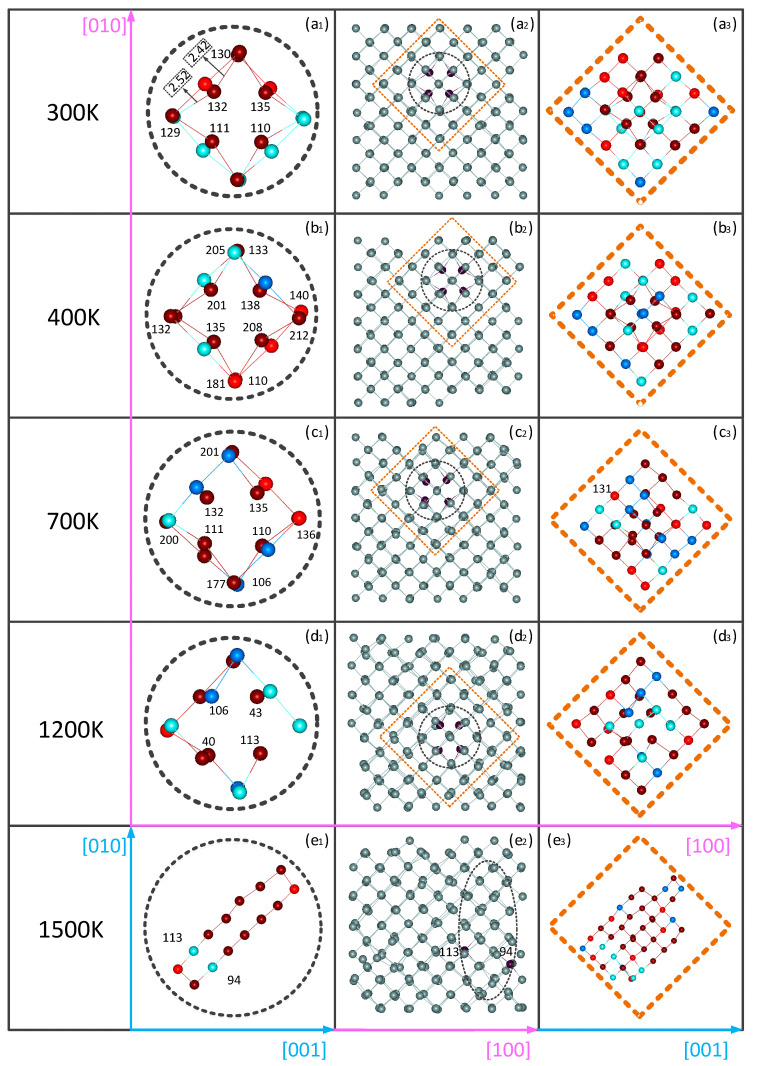
The atomic packing structure and the pressure distribution of Si lattice with one vacancy at different temperatures: (**a_1_**–**a_3_**) 300, (**b_1_**–**b_3_**) 400, (**c_1_**–**c_3_**) 700, (**d_1_**–**d_3_**) 1200 and (**e_1_**–**e_3_**) 1500 K; the black dotted line shows the packing structure and the pressure of the nearest neighbor atoms of vacancy, while the orange dotted line shows the packing structure and the pressure of the atoms neighboring the vacancy; the red atoms in this figure indicate that the atom undergoes compression, and the pressure is positive. The atomic pressure of dark red atoms is higher than that of light red atoms. Blue atoms are stretched, and the pressure is negative. Dark blue color indicates atoms that are under less pressure than light blue atoms.

**Figure 5 materials-14-03127-f005:**
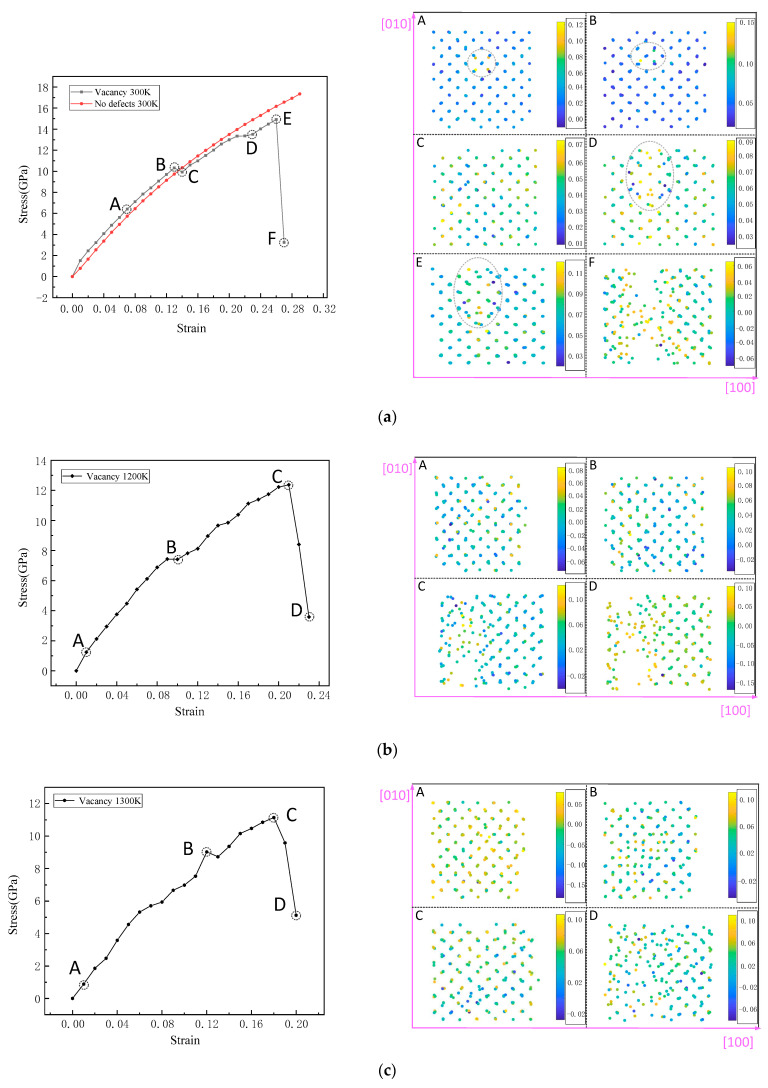
Stress varying with the strain and pressure distribution of (**a**) the prefect Si lattice and the Si lattice with one vacancy at 300 K, (**b**) the Si lattice with one vacancy at 1200 and (**c**) 1300 K along the [100] axis. The pictures on the right shows the packing structures corresponding to the letters at different strains on the left.

**Figure 6 materials-14-03127-f006:**
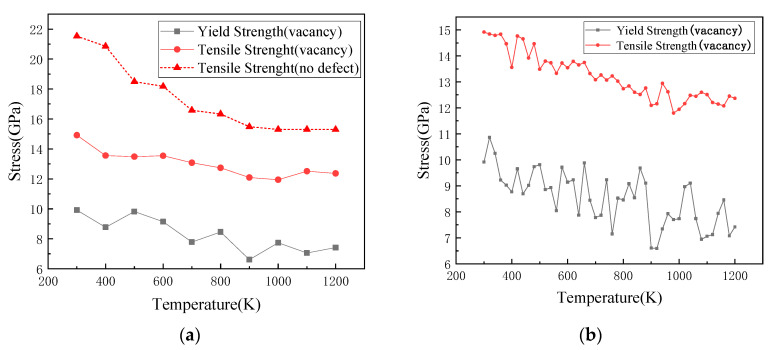
Variations of yield strength and tensile strength of the Si lattice with temperature; (**a**) a temperature increase of 100 K; and (**b**) a temperature increase of 20 K.

**Table 1 materials-14-03127-t001:** SW potential parameter.

*σ* (nm)	*ε* (kcal/mol)	*A*	*B*	*λ*	*γ*	*p*	*q*	*a*
0.20951	50	7.049556	0.602225	21.0	1.20	4	0	1.80

## Data Availability

Not applicable.

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
