# Peer review of "Atomic Simulations of Packing Structures, Local Stress and Mechanical Properties for One Silicon Lattice with Single Vacancy on Heating"

_materials, 2021, doi:10.3390/ma14113127_

Round 1

Reviewer 1 Report

My review comments can be found in the attached pdf file.

Reviewer 2 Report

Please find the review comments in attachment.

Round 2

Reviewer 1 Report

First of all, the authors have done good work to the reviewers’ comments to improve the manuscript. A few minor issues have listened below:

Page by page comments:

#1 Page 1, line 41 and following: The authors should re-write the following sentence “For example, in tensile testing,…” as this sentence is very confusing.

#2 Page 7, line 246-251: First, the sentence “Due to melting is a process…” is grammatically incorrect and has to be revised. Second, “et al.” in the following line is also incorrect and has to be fixed. Third, the sentence “For a bulk in the present simulation…” makes no logical sense and has to be improved.

#3 Conclusion, line 436-440: Please correct the errors in writing in this paragraph.

English must be still improved throughout the manuscript. Please also check all numbers and units within this realm because there must be a space between them, e.g., “100 K”.
